# Differentiable Augmentation
# for Data-Efficient GAN Training

**Shengyu Zhao**
IIIS, Tsinghua University and MIT

**Zhijian Liu**
MIT

**Ji Lin**
MIT

**Jun-Yan Zhu**
Adobe and CMU

**Song Han**
MIT

## Abstract

The performance of generative adversarial networks (GANs) heavily deteriorates given a limited amount of training data. This is mainly because the discriminator is memorizing the exact training set. To combat it, we propose *Differentiable Augmentation* (*DiffAugment*), a simple method that improves the data efficiency of GANs by imposing various types of differentiable augmentations on both real and fake samples. Previous attempts to directly augment the training data manipulate the distribution of real images, yielding little benefit; DiffAugment enables us to adopt the differentiable augmentation for the generated samples, effectively stabilizes training, and leads to better convergence. Experiments demonstrate consistent gains of our method over a variety of GAN architectures and loss functions for both unconditional and class-conditional generation. With DiffAugment, we achieve a state-of-the-art FID of 6.80 with an IS of 100.8 on ImageNet 128×128 and 2-4× reductions of FID given 1,000 images on FFHQ and LSUN. Furthermore, with only 20% training data, we can match the top performance on CIFAR-10 and CIFAR-100. Finally, our method can generate high-fidelity images using only 100 images without pre-training, while being on par with existing transfer learning algorithms. Code is available at https://github.com/mit-han-lab/data-efficient-gans.

## 1  Introduction

Big data has enabled deep learning algorithms achieve rapid advancements. In particular, state-of-the-art generative adversarial networks (GANs) [11] are able to generate high-fidelity natural images of diverse categories [2, 18]. Many computer vision and graphics applications have been enabled [32, 43, 53]. However, this success comes at the cost of a tremendous amount of computation and data. Recently, researchers have proposed promising techniques to improve the *computational efficiency* of model inference [22, 36], while the *data efficiency* remains to be a fundamental challenge.

GANs heavily rely on vast quantities of diverse and high-quality training examples. To name a few, the FFHQ dataset [17] contains 70,000 selective post-processed high-resolution images of human faces; the ImageNet dataset [6] annotates more than a million of images with various object categories. Collecting such large-scale datasets requires *months or even years* of considerable human efforts along with prohibitive annotation costs. In some cases, it is not even possible to have that many examples, *e.g.*, images of rare species or photos of a specific person or landmark. Thus, it is of critical importance to eliminate the need of immense datasets for GAN training. However, reducing the amount of training data results in drastic degradation in the performance. For example in Figure 1, given only 10% or 20% of the CIFAR-10 data, the training accuracy of the discriminator saturates quickly (to nearly 100%); however, its validation accuracy keeps decreasing (to lower than 30%), suggesting that the discriminator is simply memorizing the entire training set. This severe over-fitting problem disrupts the training dynamics and leads to degraded image quality.

A widely-used strategy to reduce overfitting in image classification is data augmentation [20, 38, 42], which can increase the diversity of training data without collecting new samples. Transformations such

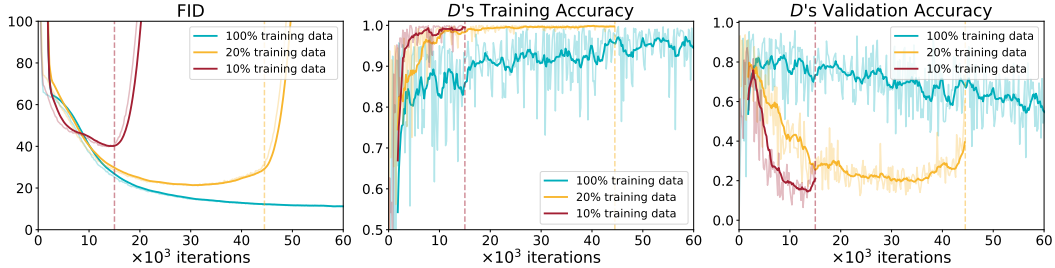

Figure 1: **BigGAN heavily deteriorates given a limited amount of data.** *left*: With 10% of CIFAR-10 data, FID increases shortly after the training starts, and the model then collapses (red curve). *middle*: the training accuracy of the discriminator $D$ quickly saturates. *right*: the validation accuracy of $D$ dramatically falls, indicating that $D$ has memorized the exact training set and fails to generalize.

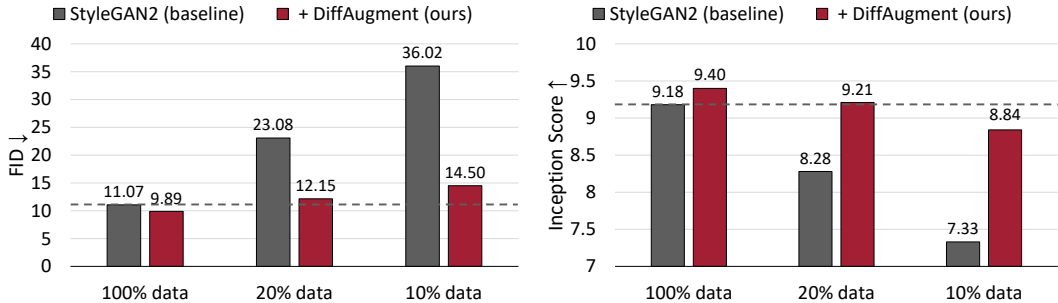

Figure 2: **Unconditional generation results on CIFAR-10.** StyleGAN2's performance drastically degrades given less training data. With DiffAugment, we are able to roughly match its FID and outperform its Inception Score (IS) using only 20% training data. FID and IS are measured using 10k samples; the validation set is used as the reference distribution for FID calculation.

as cropping, flipping, scaling, color jittering [20], and region masking (Cutout) [8] are commonly-used augmentations for vision models. However, applying data augmentation to GANs is fundamentally different. If the transformation is only added to the real images, the generator would be encouraged to match the distribution of the *augmented* images. As a consequence, the outputs suffer from distribution shift and the introduced artifacts (*e.g.*, a region being masked, unnatural color, see Figure 5a). Alternatively, we can augment both the real and generated images when training the discriminator; however, this would break the subtle balance between the generator and discriminator, leading to poor convergence as they are optimizing completely different objectives (see Figure 5b).

To combat it, we introduce a simple but effective method, *DiffAugment*, which applies the same differentiable augmentation to both real and fake images for both generator and discriminator training. It enables the gradients to be propagated through the augmentation back to the generator, regularizes the discriminator without manipulating the target distribution, and maintains the balance of training dynamics. Experiments on a variety of GAN architectures and datasets consistently demonstrate the effectiveness of our method. With DiffAugment, we improve BigGAN and achieve a Fréchet Inception Distance (FID) of 6.80 with an Inception Score (IS) of 100.8 on ImageNet 128×128 without the truncation trick [2] and reduce the StyleGAN2 baseline's FID by 2-4× given 1,000 images on the FFHQ and LSUN datasets. We also match the top performance on CIFAR-10 and CIFAR-100 using only 20% training data (see Figure 2). Furthermore, our method can generate high-quality images with only 100 examples (see Figure 3). Without any pre-training, we achieve competitive performance with existing transfer learning algorithms that used to require tens of thousands of training images.

## 2   Related Work

**Generative Adversarial Networks.**   Following the pioneering work of GAN [11], researchers have explored different ways to improve its performance and training stability. Recent efforts are centered on more stable objective functions [1, 12, 26, 27, 35], more advanced architectures [28, 29, 33, 48],

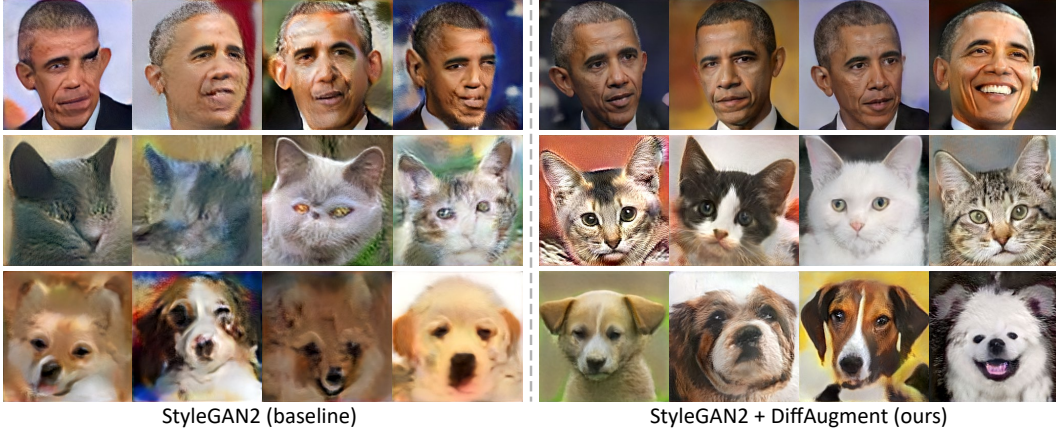

StyleGAN2 (baseline)    StyleGAN2 + DiffAugment (ours)

Figure 3: **Low-shot generation without pre-training.** Our method can generate high-fidelity images using only 100 Obama portraits (top) from our collected 100-shot datasets, 160 cats (middle) or 389 dogs (bottom) from the AnimalFace dataset [37] at 256×256 resolution. See Figure 7 for the interpolation results; nearest neighbor tests are provided in the supplementary material.

and better training strategy [7, 15, 24, 49]. As a result, both the visual fidelity and diversity of generated images have increased significantly. For example, BigGAN [2] is able to synthesize natural images with a wide range of object classes at high resolution, and StyleGAN [17, 18] can produce photorealistic face portraits with large varieties, often indistinguishable from natural photos. However, the above work paid less attention to the data efficiency aspect. A recent attempt [3, 25] leverages semi- and self-supervised learning to reduce the amount of human annotation required for training. In this paper, we study a more challenging scenario where both data and labels are limited.

**Regularization for GANs.**    GAN training often requires additional regularization as they are highly unstable. To stabilize the training dynamics, researchers have proposed several techniques including the instance noise [39], Jensen-Shannon regularization [34], gradient penalties [12, 27], spectral normalization [28], adversarial defense regularization [52], and consistency regularization [50]. All of these regularization techniques implicitly or explicitly penalize sudden changes in the discriminator's output within a local region of the input. In this paper, we provide a different perspective, data augmentation, and we encourage the discriminator to perform well under different types of augmentation. In Section 4, we show that our method is complementary to the regularization techniques in practice.

**Data Augmentation.**    Many deep learning models adopt label-preserving transformations to reduce overfitting: *e.g.*, color jittering [20], region masking [8], flipping, rotation, cropping [20, 42], data mixing [47], and local and affine distortion [38]. Recently, AutoML [40, 54] has been used to explore adaptive augmentation policies for a given dataset and task [4, 5, 23]. However, applying data augmentation to generative models, such as GANs, remains an open question. Different from the classifier training where the label is invariant to transformations of the input, the goal of generative models is to learn the data distribution itself. Directly applying augmentation would inevitably alter the distribution. We present a simple strategy to circumvent the above concern. Concurrent with our work, several methods [16, 41, 51] independently proposed data augmentation for training GANs. We urge the readers to check out their work for more details.

## 3    Method

Generative adversarial networks (GANs) aim to model the distribution of a target dataset via a generator $G$ and a discriminator $D$. The generator $G$ maps an input latent vector $z$, typically drawn from a Gaussian distribution, to its output $G(z)$. The discriminator $D$ learns to distinguish generated samples $G(z)$ from real observations $x$. The standard GANs training algorithm alternately optimizes the discriminator's loss $L_D$ and the generator's loss $L_G$ given loss functions $f_D$ and $f_G$:

$$L_D = \mathbb{E}_{x \sim p_{\text{data}}(x)}[f_D(-D(x))] + \mathbb{E}_{z \sim p(z)}[f_D(D(G(z)))], \tag{1}$$

$$L_G = \mathbb{E}_{z \sim p(z)}[f_G(-D(G(z)))]. \tag{2}$$

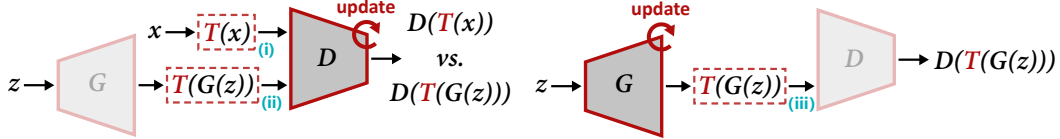

Figure 4: **Overview of DiffAugment** for updating $D$ (left) and $G$ (right). DiffAugment applies the augmentation $T$ to both the real samples $x$ and the generated output $G(z)$. When we update $G$, gradients need to be back-propagated through $T$, which requires $T$ to be differentiable w.r.t. the input.

| Method | Where $T$? | | | Color + Transl. + Cutout | | Transl. + Cutout | | Translation | |
|---|---|---|---|---|---|---|---|---|---|
| | (i) | (ii) | (iii) | IS | FID | IS | FID | IS | FID |
| BigGAN (baseline) | | | | 9.06 | 9.59 | 9.06 | 9.59 | 9.06 | 9.59 |
| Aug. reals only | ✓ | | | 5.94 | 49.38 | 6.51 | 37.95 | 8.40 | 19.16 |
| Aug. reals + fakes ($D$ only) | ✓ | ✓ | | 3.00 | 126.96 | 3.76 | 114.14 | 3.50 | 100.13 |
| DiffAugment ($D + G$, ours) | ✓ | ✓ | ✓ | **9.25** | **8.59** | **9.16** | **8.70** | **9.07** | **9.04** |

Table 1: **DiffAugment vs. vanilla augmentation strategies** on CIFAR-10 with 100% training data. "Augment reals only" applies augmentation $T$ to (i) only (see Figure 4) and corresponds to Equations (3)-(4); "Augment $D$ only" applies $T$ to both reals (i) and fakes (ii), but not $G$ (iii), and corresponds to Equations (5)-(6); "DiffAugment" applies $T$ to reals (i), fakes (ii), and $G$ (iii). (iii) requires $T$ to be differentiable since gradients should be back-propagated through $T$ to $G$. DiffAugment corresponds to Equations (7)-(8). IS and FID are measured using 10k samples; the validation set is the reference distribution. We select the snapshot with the best FID for each method. Results are averaged over 5 evaluation runs; all standard deviations are less than 1% relatively.

Here, different loss functions can be used, such as the non-saturating loss [11], where $f_D(x) = f_G(x) = \log{(1 + e^x)}$, and the hinge loss [28], where $f_D(x) = \max(0, 1 + x)$ and $f_G(x) = x$.

Despite extensive ongoing efforts of better GAN architectures and loss functions, a fundamental challenge still exists: the discriminator tends to *memorize* the observations as the training progresses. An overfitted discriminator penalizes any generated samples other than the exact training data points, provides uninformative gradients due to poor generalization, and usually leads to training instability.

**Challenge: Discriminator Overfitting.** Here we analyze the performance of BigGAN [2] with different amounts of data on CIFAR-10. As plotted in Figure 1, even given 100% data, the gap between the discriminator's training and validation accuracy keeps increasing, suggesting that the discriminator is simply memorizing the training images. This happens not only on limited data but also on the large-scale ImageNet dataset, as observed by Brock *et al*. [2]. BigGAN already adopts Spectral Normalization [28], a widely-used regularization technique for both generator and discriminator architectures, but still suffers from severe overfitting.

## 3.1 Revisiting Data Augmentation

Data augmentation is a commonly-used strategy to reduce overfitting in many recognition tasks — it has an irreplaceable role and can also be applied in conjunction with other regularization techniques: *e.g.*, weight decay. We have shown that the discriminator suffers from a similar overfitting problem as the binary classifier. However, data augmentation is seldom used in the GAN literature compared to the explicit regularizations on the discriminator [12, 27, 28]. In fact, a recent work [50] observes that directly applying data augmentation to GANs does not improve the baseline. So, we would like to ask the questions: what prevents us from simply applying data augmentation to GANs? Why is augmenting GANs not as effective as augmenting classifiers?

**Augment reals only.** The most straightforward way of augmenting GANs would be directly applying augmentation $T$ to the real observations $x$, which we call "Augment reals only":

$$L_D = \mathbb{E}_{x \sim p_{\text{data}}(x)}[f_D(-D(T(x)))] + \mathbb{E}_{z \sim p(z)}[f_D(D(G(z)))], \quad (3)$$

$$L_G = \mathbb{E}_{z \sim p(z)}[f_G(-D(G(z)))]. \quad (4)$$

However, "Augment reals only" deviates from the original purpose of generative modeling, as the model is now learning a different data distribution of $T(x)$ instead of $x$. This prevents us from

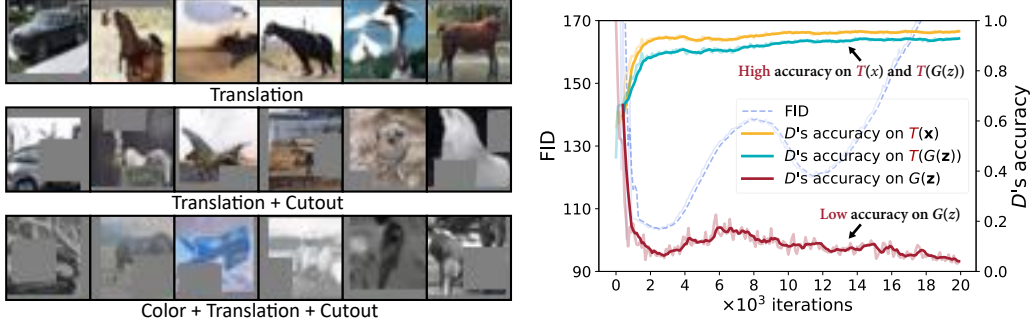

(a) "Augment reals only": the same augmentation artifacts appear on the generated images.

(b) "Augment $D$ only": the unbalanced optimization between $G$ and $D$ cripples training.

Figure 5: **Understanding why vanilla augmentation strategies fail:** (a) "Augment reals only" mimics the same data distortion as introduced by the augmentations, *e.g.*, the translation padding, the Cutout square, and the color artifacts; (b) "Augment $D$ only" diverges because of the unbalanced optimization — $D$ perfectly classifies the augmented images (both $T(\boldsymbol{x})$ and $T(G(\boldsymbol{z}))$ but barely recognizes $G(\boldsymbol{z})$ (i.e., fake images without augmentation) from which $G$ receives gradients.

applying any augmentation that significantly alters the distribution of the real images. The choices that meet this requirement, although strongly dependent on the specific dataset, can only be horizontal flips in most cases. We find that applying random horizontal flips does increase the performance moderately, and we use it in all our experiments to make our baselines stronger. We demonstrate the side effects of enforcing stronger augmentations quantitatively in Table 1 and qualitatively in Figure 5a. As expected, the model learns to produce unwanted color and geometric distortion (*e.g.*, unnatural color, cutout holes) as introduced by these augmentations, resulting in a significantly worse performance (see "Augment reals only" in Table 1).

**Augment $D$ only.** Previously, "Augment reals only" applies one-sided augmentation to the real samples, and hence the convergence can be achieved only if the generated distribution matches the manipulated real distribution. From the discriminator's perspective, it may be tempting to augment both real and fake samples when we update $D$:

$$L_D = \mathbb{E}_{\boldsymbol{x} \sim p_{\text{data}}(\boldsymbol{x})}[f_D(-D(T(\boldsymbol{x})))] + \mathbb{E}_{\boldsymbol{z} \sim p(\boldsymbol{z})}[f_D(D(T(G(\boldsymbol{z}))))], \tag{5}$$

$$L_G = \mathbb{E}_{\boldsymbol{z} \sim p(\boldsymbol{z})}[f_G(-D(G(\boldsymbol{z})))]. \tag{6}$$

Here, the same function $T$ is applied to both real samples $\boldsymbol{x}$ and fake samples $G(\boldsymbol{z})$. If the generator successfully models the distribution of $\boldsymbol{x}$, $T(G(\boldsymbol{z}))$ and $T(\boldsymbol{x})$ should be indistinguishable to the discriminator as well as $G(\boldsymbol{z})$ and $\boldsymbol{x}$. However, this strategy leads to even worse results (see "Augment $D$ only" in Table 1). Figure 5b plots the training dynamics of "Augment $D$ only" with *Translation* applied. Although $D$ classifies the augmented images (both $T(G(\boldsymbol{z}))$ and $T(\boldsymbol{x})$) perfectly with an accuracy of above 90%, it fails to recognize $G(\boldsymbol{z})$, the generated images without augmentation, with an accuracy of lower than 10%. As a result, the generator completely fools the discriminator by $G(\boldsymbol{z})$ and cannot obtain useful information from the discriminator. This suggests that any attempts that break the delicate balance between the generator $G$ and discriminator $D$ are prone to failure.

### 3.2 Differentiable Augmentation for GANs

The failure of "Augment reals only" motivates us to augment both *real* and *fake* samples, while the failure of "Augment $D$ only" warns us that the *generator* should not neglect the augmented samples. Therefore, to propagate gradients through the augmented samples to $G$, the augmentation $T$ must be *differentiable* as depicted in Figure 4. We call this *Differentiable Augmentation (DiffAugment)*:

$$L_D = \mathbb{E}_{\boldsymbol{x} \sim p_{\text{data}}}(\boldsymbol{x})[f_D(-D(T(\boldsymbol{x})))] + \mathbb{E}_{\boldsymbol{z} \sim p(\boldsymbol{z})}[f_D(D(T(G(\boldsymbol{z}))))], \tag{7}$$

$$L_G = \mathbb{E}_{\boldsymbol{z} \sim p(\boldsymbol{z})}[f_G(-D(T(G(\boldsymbol{z}))))]. \tag{8}$$

Note that $T$ is required to be the same (random) function but not necessarily the same random seed across the three places illustrated in Figure 4. We demonstrate the effectiveness of DiffAugment using three simple choices of transformations and its composition, throughout the paper: *Translation*

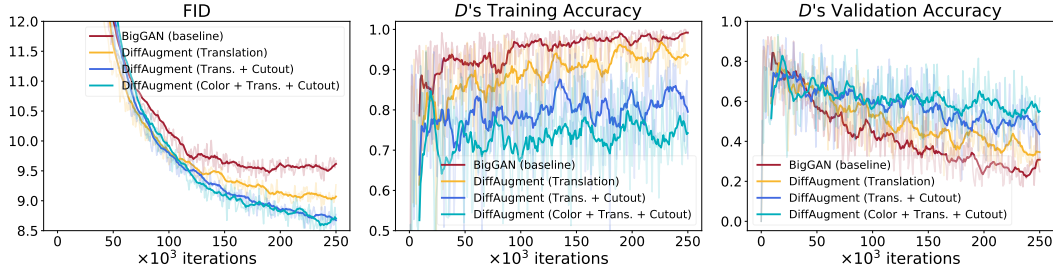

Figure 6: **Analysis of different types of DiffAugment** on CIFAR-10 with 100% training data. A stronger DiffAugment can dramatically reduce the gap between the discriminator's training accuracy (middle) and validation accuracy (right), leading to a better convergence (left).

| Method | 100% training data | | 50% training data | | 25% training data | |
|---|---|---|---|---|---|---|
| | IS | FID | IS | FID | IS | FID |
| BigGAN [2] | $94.5 \pm 0.4$ | $7.62 \pm 0.02$ | $89.9 \pm 0.2$ | $9.64 \pm 0.04$ | $46.5 \pm 0.4$ | $25.37 \pm 0.07$ |
| + DiffAugment | $\mathbf{100.8 \pm 0.2}$ | $\mathbf{6.80 \pm 0.02}$ | $\mathbf{91.9 \pm 0.5}$ | $\mathbf{8.88 \pm 0.06}$ | $\mathbf{74.2 \pm 0.5}$ | $\mathbf{13.28 \pm 0.07}$ |

Table 2: **ImageNet** $128 \times 128$ results without the truncation trick [2]. IS and FID are measured using 50k samples; the validation set is used as the reference distribution for FID. We select the snapshot with the best FID for each method. We report means and standard deviations over 3 evaluation runs.

(within $[-1/8, 1/8]$ of the image size, padded with zeros), *Cutout* [8] (masking with a random square of half image size), and *Color* (including random brightness within $[-0.5, 0.5]$, contrast within $[0.5, 1.5]$, and saturation within $[0, 2]$). As shown in Table 1, BigGAN can be improved using the simple *Translation* policy and further boosted using a composition of *Cutout* and *Translation*; it is also robust to the strongest policy when *Color* is used in combined. Figure 6 analyzes that stronger DiffAugment policies generally maintain a higher discriminator's validation accuracy at the cost of a lower training accuracy, alleviate the overfitting problem, and eventually achieve better convergence.

# 4 Experiments

We conduct extensive experiments on ImageNet [6], CIFAR-10 [19], CIFAR-100, FFHQ [17], and LSUN-Cat [46] based on the leading class-conditional BigGAN [2] and unconditional Style-GAN2 [18]. We use the common evaluation metrics Fréchet Inception Distance (FID) [13] and Inception Score (IS) [35]. In addition, we apply our method to low-shot generation both with and without pre-training in Section 4.4. Finally, we perform analysis studies in Section 4.5.

## 4.1 ImageNet

We follow the top-performing model BigGAN [2] on ImageNet dataset at $128 \times 128$ resolution. Additionally, we augment real images with random horizontal flips, yielding the best reimplementation of BigGAN to our knowledge (FID: ours 7.6 *vs.* 8.7 in the original paper [2]). We use the simple *Translation* DiffAugment for all the data percentage settings. In Table 2, our method achieves significant gains especially under the 25% data setting, in which the baseline model undergoes an early collapse, and advances the state-of-the-art FID and IS with 100% data available.

## 4.2 FFHQ and LSUN-Cat

We further experiment with StyleGAN2 [18] on the FFHQ portrait dataset [17] and the LSUN-Cat dataset [46] at $256 \times 256$ resolution. We investigate different limited data settings, with 1k, 5k, 10k, and 30k training images available. We apply the strongest *Color + Translation + Cutout* DiffAugment to all the StyleGAN2 baselines without any hyperparameter changes. The real images are also augmented with random horizontal flips as commonly applied in StyleGAN2 [18]. Results are shown in Table 3. Our performance gains are considerable under all the data percentage settings. Moreover, with the fixed policies used in DiffAugment, our performance is on par with ADA [16], a concurrent work based on the adaptive augmentation strategy.

| Method | FFHQ | | | | LSUN-Cat | | | |
|---|---|---|---|---|---|---|---|---|
| | 30k | 10k | 5k | 1k | 30k | 10k | 5k | 1k |
| ADA [16] | 5.46 | 8.13 | 10.96 | **21.29** | 10.50 | 13.13 | 16.95 | 43.25 |
| StyleGAN2 [18] | 6.16 | 14.75 | 26.60 | 62.16 | 10.12 | 17.93 | 34.69 | 182.85 |
| + DiffAugment | **5.05** | **7.86** | **10.45** | 25.66 | **9.68** | **12.07** | **16.11** | **42.26** |

Table 3: **FFHQ and LSUN-Cat** results with 1k, 5k, 10k, and 30k training samples. With the fixed *Color + Translation + Cutout* DiffAugment, our method improves the StyleGAN2 baseline and is on par with a concurrent work ADA [16]. FID is measured using 50k generated samples; the full training set is used as the reference distribution. We select the snapshot with the best FID for each method. Results are averaged over 5 evaluation runs; all standard deviations are less than 1% relatively.

| Method | CIFAR-10 | | | CIFAR-100 | | |
|---|---|---|---|---|---|---|
| | 100% data | 20% data | 10% data | 100% data | 20% data | 10% data |
| BigGAN [2] | 9.59 | 21.58 | 39.78 | 12.87 | 33.11 | 66.71 |
| + DiffAugment | **8.70** | **14.04** | **22.40** | **12.00** | **22.14** | **33.70** |
| CR-BigGAN [50] | 9.06 | 20.62 | 37.45 | 11.26 | 36.91 | 47.16 |
| + DiffAugment | **8.49** | **12.84** | **18.70** | **11.25** | **20.28** | **26.90** |
| StyleGAN2 [18] | 11.07 | 23.08 | 36.02 | 16.54 | 32.30 | 45.87 |
| + DiffAugment | **9.89** | **12.15** | **14.50** | **15.22** | **16.65** | **20.75** |

Table 4: **CIFAR-10 and CIFAR-100** results. We select the snapshot with the best FID for each method. Results are averaged over 5 evaluation runs; all standard deviations are less than 1% relatively. We use 10k samples and the validation set as the reference distribution for FID calculation, as done in prior work [50]. Concurrent works [14, 16] use a different protocol: 50k samples and the training set as the reference distribution. If we adopt this evaluation protocol, our BigGAN + DiffAugment achieves an FID of 4.61, CR-BigGAN + DiffAugment achieves an FID of 4.30, and StyleGAN2 + DiffAugment achieves an FID of 5.79.

## 4.3 CIFAR-10 and CIFAR-100

We experiment on the class-conditional BigGAN [2] and CR-BigGAN [50] and unconditional StyleGAN2 [18] models. For a fair comparison, we also augment real images with random horizontal flips for all the baselines. The baseline models already adopt advanced regularization techniques, including Spectral Normalization [28], Consistency Regularization [50], and $R_1$ regularization [27]; however, none of them achieves satisfactory results under the 10% data setting. For DiffAugment, we adopt *Translation + Cutout* for the BigGAN models, *Color + Cutout* for StyleGAN2 with 100% data, and *Color + Translation + Cutout* for StyleGAN2 with 10% or 20% data. As summarized in Table 4, our method improves all the baselines independently of the baseline architectures, regularizations, and loss functions (hinge loss in BigGAN and non-saturating loss in StyleGAN2) without any hyperparameter changes. We refer the readers to the supplementary material for the complete tables with IS. The improvements are considerable especially when limited data is available. This is, to our knowledge, the new state of the art on CIFAR-10 and CIFAR-100 for both class-conditional and unconditional generation under all the 10%, 20%, and 100% data settings.

## 4.4 Low-Shot Generation

For a certain person, an object, or a landmark, it is often tedious, if not completely impossible, to collect a large-scale dataset. To address this, researchers recently exploit few-shot learning [9, 21] in the setting of image generation. Wang *et al*. [45] use fine-tuning to transfer the knowledge of models pre-trained on external large-scale datasets. Several works propose to fine-tune only part of the model [30, 31, 44]. Below, we show that our method not only produces competitive results without using external datasets or models but also is orthogonal to the existing transfer learning methods.

We replicate the recent transfer learning algorithms [30, 31, 44, 45] using the same codebase as Mo *et al*. [30] on their datasets (AnimalFace [37] with 160 cats and 389 dogs), based on the pre-trained StyleGAN model from the FFHQ face dataset [17]. To further demonstrate the data efficiency,

| Method | Pre-training? | 100-shot | | | AnimalFace [37] | |
| --- | --- | --- | --- | --- | --- | --- |
| | | Obama | Grumpy cat | Panda | Cat | Dog |
| Scale/shift [31] | Yes | 50.72 | 34.20 | 21.38 | 54.83 | 83.04 |
| MineGAN [44] | Yes | 50.63 | 34.54 | 14.84 | 54.45 | 93.03 |
| TransferGAN [45] | Yes | 48.73 | 34.06 | 23.20 | 52.61 | 82.38 |
| + DiffAugment | Yes | **39.85** | **29.77** | **17.12** | **49.10** | **65.57** |
| FreezeD [30] | Yes | 41.87 | 31.22 | 17.95 | 47.70 | 70.46 |
| + DiffAugment | Yes | **35.75** | **29.34** | **14.50** | **46.07** | **61.03** |
| StyleGAN2 [18] | No | 80.20 | 48.90 | 34.27 | 71.71 | 130.19 |
| + DiffAugment | No | **46.87** | **27.08** | **12.06** | **42.44** | **58.85** |

Table 5: **Low-shot generation** results. With only **100** (Obama, Grumpy cat, Panda), **160** (Cat), or **389** (Dog) training images, our method is on par with the transfer learning algorithms that are pre-trained with **70,000** images. FID is measured using 5k generated samples; the training set is the reference distribution. We select the snapshot with the best FID for each method.

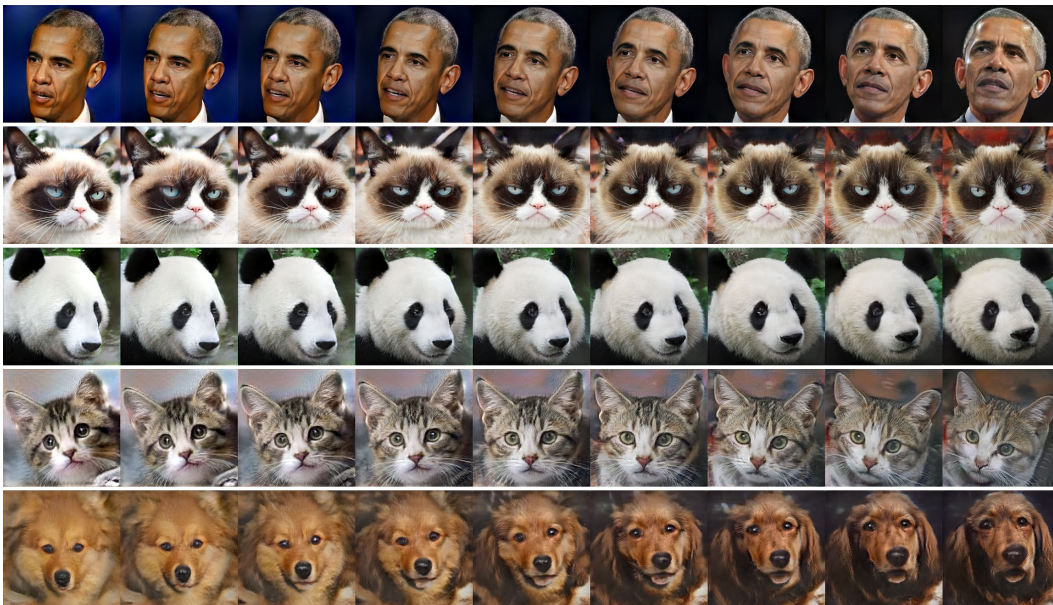

Figure 7: **Style space interpolation** of our method for low-shot generation without pre-training. The smooth interpolation results suggest little overfitting of our method even given small datasets.

we collect the 100-shot Obama, grumpy cat, and panda datasets, and train the StyleGAN2 model on each dataset using only 100 images without pre-training. For DiffAugment, we adopt *Color + Translation + Cutout* for StyleGAN2, *Color + Cutout* for both the vanilla fine-tuning algorithm TransferGAN [45] and FreezeD [30] that freezes the first several layers of the discriminator. Table 5 shows that DiffAugment achieves consistent gains independently of the training algorithm on all the datasets. Without any pre-training, we still achieve results on par with the existing transfer learning algorithms that require tens of thousands of images, with an exception on the 100-shot Obama dataset where pre-training with human faces clearly leads to better generalization. See Figure 3 and the supplementary material for qualitative comparisons. While there might be a concern that the generator is likely to overfit the tiny datasets (*i.e.*, generating identical training images), Figure 7 suggests little overfitting of our method via linear interpolation in the style space [17]; please refer to the supplementary material for the nearest neighbor tests.

### 4.5 Analysis

Below, we investigate whether smaller model or stronger regularization would similarly reduce overfitting and whether DiffAugment still helps. Finally, we analyze additional choices of DiffAugment.

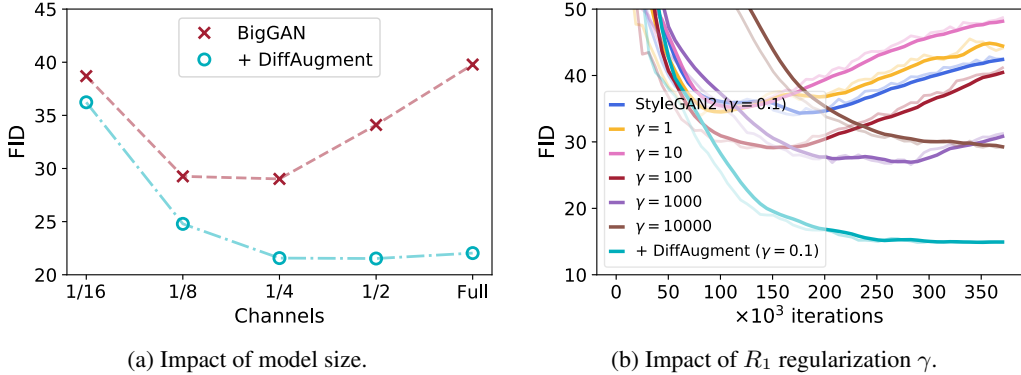

(a) Impact of model size.

(b) Impact of $R_1$ regularization $\gamma$.

Figure 8: **Analysis of smaller models or stronger regularization** on CIFAR-10 with 10% training data. (a) Smaller models reduce overfitting for the BigGAN baseline, while our method dominates its performance at all model capacities. (b) Over a wide sweep of the $R_1$ regularization $\gamma$ for the baseline StyleGAN2, its best FID (26.87) is still much worse than ours (14.50).

**Model Size Matters?** We reduce the model capacity of BigGAN by progressively halving the number of channels for both $G$ and $D$. As plotted in Figure 8a, the baseline heavily overfits on CIFAR-10 with 10% training data when using the full model and achieves a minimum FID of 29.02 at $1/4$ channels. However, it is surpassed by our method over all model capacities. With $1/4$ channels, our model achieves a significantly better FID of 21.57, while the gap is monotonically increasing as the model becomes larger. We refer the readers to the supplementary material for the IS plot.

**Stronger Regularization Matters?** As StyleGAN2 adopts the $R_1$ regularization [27] to stabilize training, we increase its strength from $\gamma = 0.1$ to up to $10^4$ and plot the FID curves in Figure 8b. While we initially find that $\gamma = 0.1$ works best under the 100% data setting, the choice of $\gamma = 10^3$ boosts its performance from 34.05 to 26.87 under the 10% data setting. When $\gamma = 10^4$, within 750k iterations, we only observe a minimum FID of 29.14 at 440k iteration and the performance deteriorates after that. However, its best FID is still **1.8×** worse than ours (with the default $\gamma = 0.1$). This shows that DiffAugment is more effective compared to explicitly regularizing the discriminator.

**Choice of DiffAugment Matters?** We investigate additional choices of DiffAugment in Figure 9, including random 90° rotations ($\{-90°, 0°, 90°\}$ each with $1/3$ probability), Gaussian noise (with a standard deviation of 0.1), and general geometry transformations that involve bilinear interpolation, such as bilinear translation (within $[-0.25, 0.25]$), bilinear scaling (within $[0.75, 1.25]$), bilinear rotation (within $[-30°, 30°]$), and bilinear shearing (within $[-0.25, 0.25]$). While all these policies consistently outperform the baseline, we find that the *Color + Translation + Cutout* DiffAugment is especially effective. The simplicity also makes it easier to deploy.

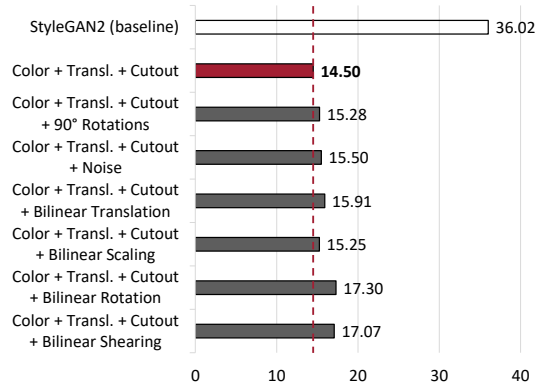

Figure 9: Various types of DiffAugment consistently outperform the baseline. We report StyleGAN2's FID on CIFAR-10 with 10% training data.

## 5 Conclusion

We present *DiffAugment* for data-efficient GAN training. DiffAugment reveals valuable observations that augmenting both real and fake samples effectively prevents the discriminator from over-fitting, and that the augmentation must be differentiable to enable both generator and discriminator training. Extensive experiments consistently demonstrate its benefits with different network architectures (StyleGAN2 and BigGAN), supervision settings, and objective functions, across multiple datasets (ImageNet, CIFAR, FFHQ, LSUN, and 100-shot datasets). Our method is especially effective when limited data is available. Our code, datasets, and models are available for future comparisons.

## Broader Impact

In this paper, we investigate GANs from the data efficiency perspective, aiming to make generative modeling accessible to more people (e.g., visual artists and novice users) and research fields who have no access to abundant data. In the real-world scenarios, there could be various reasons that lead to limited amount of data available, such as rare incidents, privacy concerns, and historical visual data [10]. DiffAugment provides a promising way to alleviate the above issues and make AI more accessible to everyone.

## Acknowledgments

We thank NSF Career Award #1943349, MIT-IBM Watson AI Lab, Google, Adobe, and Sony for supporting this research. Research supported with Cloud TPUs from Google's TensorFlow Research Cloud (TFRC). We thank William S. Peebles and Yijun Li for helpful comments.

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
