[Supplementary Material]

# Differentiable Augmentation
# for Data-Efficient GAN Training
# Supplementary Material

**Shengyu Zhao**
IIIS, Tsinghua University and MIT

**Zhijian Liu**
MIT

**Ji Lin**
MIT

**Jun-Yan Zhu**
Adobe and CMU

**Song Han**
MIT

## Appendix A    Hyperparameters and Training Details

### A.1    ImageNet Experiments

The Compare GAN codebase[*] suffices to replicate BigGAN's FID on ImageNet dataset at $128\times128$ resolution but has some small differences to the original paper [2]. First, the codebase uses a learning rate of $10^{-4}$ for $G$ and $5 \times 10^{-4}$ for $D$. Second, it processes the raw images into $128\times128$ resolution with random scaling and random cropping. Since we find that random cropping leads to a worse IS, we process the raw images with random scaling and center cropping instead. We additionally augment the images with random horizontal flips, yielding the best re-implementation of BigGAN to our knowledge. With DiffAugment, we find that $D$'s learning rate of $5 \times 10^{-4}$ often makes $D$'s loss stuck at a high level, so we reduce $D$'s learning rate to $4 \times 10^{-4}$ for the 100% data setting and $2 \times 10^{-4}$ for the 10% and 20% data settings. However, we note that the baseline model does not benefit from this reduced learning rate: if we reduce $D$'s learning rate from $5 \times 10^{-4}$ to $2 \times 10^{-4}$ under the 50% data setting, its performance degrades from an FID/IS of 9.64/89.9 to 10.79/75.7. All the models achieve the best FID within 200k iterations and deteriorate after that, taking up to 3 days on a TPU v2/v3 Pod with 128 cores.

See Figure 3 for a qualitative comparison between BigGAN and BigGAN + DiffAugment. Our method improves the image quality of the samples in both 25% and 100% data settings. The visual difference is more clear under the 25% data setting.

**Notes on CR-BigGAN [50].**    CR-BigGAN [50] reports an FID of 6.66, which is slightly better than ours 6.80 (BigGAN + DiffAugment) with 100% data. However, the code and pre-trained models of CR-BigGAN [50] are not available, while its IS is not reported either. Our reimplemented CR-BigGAN only achieves an FID of 7.95 with an IS of 82.0, even worse than the baseline BigGAN. Nevertheless, our CIFAR experiments suggest the potential of applying DiffAugment on top of CR.

### A.2    FFHQ and LSUN-Cat Experiments

We use the official TensorFlow implementation of StyleGAN2[†] and the default network configuration at $256\times256$ resolution with an $R_1$ regularization $\gamma$ of 1, but without the path length regularization and the lazy regularization since they do not improve FID [18]. The number of feature maps at shallow layers ($64\times64$ resolution and above) is halved to match the architecture of ADA [16]. All the models in our experiments are augmented with random horizontal flips, trained on 8 GPUs with a maximum training length of 25,000k images.

See Figure 4-5 for qualitative comparisons between StyleGAN2 and StyleGAN2 + DiffAugment. Our method considerably improves the image quality with limited data available.

---

[*]https://github.com/google/compare_gan
[†]https://github.com/NVlabs/stylegan2

Figure 1: Unconditional generation results on **CIFAR-100**. We are able to roughly match Style-GAN2's FID and outperform its IS using only 20% training data.

Figure 2: **Analysis of smaller models or stronger regularization** on CIFAR-10 with **10%** training data. *left*: Smaller models reduce overfitting for the BigGAN baseline, while our method outperforms it at all the model capacities. *right*: Over a wide sweep of the $R_1$ regularization $\gamma$ for the baseline StyleGAN2, its best IS (7.75) is still **12%** worse than ours (8.84).

| Method | 100% training data | | 20% training data | | 10% training data | |
|---|---|---|---|---|---|---|
| | IS | FID | IS | FID | IS | FID |
| BigGAN [2] | 9.06 | 9.59 | 8.41 | 21.58 | 7.62 | 39.78 |
| + DiffAugment | **9.16** | **8.70** | **8.65** | **14.04** | **8.09** | **22.40** |
| CR-BigGAN [50] | **9.20** | 9.06 | 8.43 | 20.62 | 7.66 | 37.45 |
| + DiffAugment | 9.17 | **8.49** | **8.61** | **12.84** | **8.49** | **18.70** |
| StyleGAN2 [18] | 9.18 | 11.07 | 8.28 | 23.08 | 7.33 | 36.02 |
| + DiffAugment | **9.40** | **9.89** | **9.21** | **12.15** | **8.84** | **14.50** |

Table 1: **CIFAR-10 results.** IS and FID are measured using 10k samples; the validation set is the reference distribution for FID calculation. We select the snapshot with the best FID for each method. Results are averaged over 5 evaluation runs; all standard deviations are less than 1% relatively.

| Method | 100% training data | | 20% training data | | 10% training data | |
|---|---|---|---|---|---|---|
| | IS | FID | IS | FID | IS | FID |
| BigGAN [2] | **10.92** | 12.87 | 9.11 | 33.11 | 5.94 | 66.71 |
| + DiffAugment | 10.66 | **12.00** | **9.47** | **22.14** | **8.38** | **33.70** |
| CR-BigGAN [50] | **10.95** | 11.26 | 8.44 | 36.91 | 7.91 | 47.16 |
| + DiffAugment | 10.81 | **11.25** | **9.12** | **20.28** | **8.70** | **26.90** |
| StyleGAN2 [18] | 9.51 | 16.54 | 7.86 | 32.30 | 7.01 | 45.87 |
| + DiffAugment | **10.04** | **15.22** | **9.82** | **16.65** | **9.06** | **20.75** |

Table 2: **CIFAR-100 results.** IS and FID are measured using 10k samples; the validation set is the reference distribution for FID calculation. We select the snapshot with the best FID for each method. Results are averaged over 5 evaluation runs; all standard deviations are less than 1% relatively.

## A.3   CIFAR-10 and CIFAR-100 Experiments

We replicate BigGAN and CR-BigGAN baselines on CIFAR using the PyTorch implementation[‡]. All hyperparameters are kept unchanged from the default CIFAR-10 configuration, including the batch size (50), the number of $D$ steps (4) per $G$ step, and a learning rate of $2 \times 10^{-4}$ for both $G$ and $D$. The hyperparameter $\lambda$ of Consistency Regularization (CR) is set to 10 as recommended [50]. All the models are run on 2 GPUs with a maximum of 250k training iterations on CIFAR-10 and 500k iterations on CIFAR-100.

For StyleGAN2, we use the official TensorFlow implementation[§] but include some changes to make it work better on CIFAR. The number of channels is 128 at 32×32 resolution and doubled at each coarser level with a maximum of 512 channels. We set the half-life of the exponential moving average of the generator's weights to 1,000k instead of 10k images since it stabilizes the FID curve and leads to consistently better performance. We set $\gamma = 0.1$ instead of 10 for the $R_1$ regularization, which significantly improves the baseline's performance under the 100% data setting on CIFAR. The path length regularization and the lazy regularization are also disabled. The baseline model can already achieve the best FID and IS to our knowledge for unconditional generation on the CIFAR datasets. All StyleGAN2 models are trained on 4 GPUs with the default batch size (32) and a maximum training length of 25,000k images.

We apply DiffAugment to BigGAN, CR-BigGAN, and StyleGAN2 without changes to the baseline settings. There are several things to note when applying DiffAugment in conjunction with gradient penalties [12] or CR [50]. The $R_1$ regularization penalizes the gradients of $D(\boldsymbol{x})$ w.r.t. the input $\boldsymbol{x}$. With DiffAugment, the gradients of $D(T(\boldsymbol{x}))$ can be calculated w.r.t. either $\boldsymbol{x}$ or $T(\boldsymbol{x})$. We choose to penalize the gradients of $D(T(\boldsymbol{x}))$ w.r.t. $T(\boldsymbol{x})$ for the CIFAR, FFHQ, and LSUN experiments since it slightly outperforms the other choice in practice; for the low-shot generation experiments, we penalize the gradients of $D(T(\boldsymbol{x}))$ w.r.t. $\boldsymbol{x}$ instead from which we observe better diversity of the generated images. As CR has already used image translation to calculate the consistency loss, we only apply *Cutout* DiffAugment on top of CR under the 100% data setting. For the 10% and 20% data settings, we exploit stronger regularization by directly applying CR between $\boldsymbol{x}$ and $T(\boldsymbol{x})$, i.e., before and after the *Translation + Cutout* DiffAugment.

We match the top performance for unconditional generation on CIFAR-100 as well as CIFAR-10 using only 20% data (see Figure 1). See Figure 2 for the analysis of smaller models or stronger regularization in terms of IS. See Table 1 and Table 2 for quantitative results.

## A.4   Low-Shot Generation Experiments

We compare our method to transfer learning algorithms using the FreezeD's codebase[¶] (for Transfer-GAN [45], Scale/shift [31], and FreezeD [30]) and the newly released MineGAN [44] code[‖]. All the models are transferred from a pre-trained StyleGAN model from the FFHQ dataset [17] at 256×256 resolution. FreezeD reports the best performance when freezing the first 4 layers of $D$ [30]; when applying DiffAugment to FreezeD, we only freeze the first 2 layers of $D$. All other hyperparameters are kept unchanged from the default settings. All the models are trained on 1 GPU with a maximum of 10k training iterations on our 100-shot datasets and 20k iterations on the AnimalFace [37] datasets.

When training the StyleGAN2 model from scratch, we use their default network configuration at 256×256 resolution with an $R_1$ regularization $\gamma$ of 10 but without the path length regularization and the lazy regularization. We use a smaller batch size of 16, which improves the performance of both the StyleGAN2 baseline and ours, compared to the default batch size of 32. All the models are trained on 4 GPUs with a maximum training length of 300k images on our 100-shot datasets and 500k images on the AnimalFace datasets.

See Figure 6 for the additional interpolation results, Figure 7 and Figure 8 for the nearest neighbor tests of our method without pre-training both in pixel space and in the LPIPIS feature space [51]. See Figures 9-13 for qualitative comparisons.

---

[‡] https://github.com/ajbrock/BigGAN-PyTorch
[§] https://github.com/NVlabs/stylegan2
[¶] https://github.com/sangwoomo/FreezeD
[‖] https://github.com/yaxingwang/MineGAN

## Appendix B    Evaluation Metrics

We measure FID and IS using the official Inception v3 model in TensorFlow for all the methods and datasets. Note that some papers using PyTorch implementations, including FreezeD [30], report different numbers from the official TensorFlow implementation of FID and IS. On ImageNet, CIFAR-10, and CIFAR-100, we inherit the setting from the Compare GAN codebase that the number of samples of generated images equals the number of real images in the validation set, and the validation set is used as the reference distribution for FID calculation. For the low-shot generation experiments, we sample 5k generated images and we use the training set as the reference distribution. For the FFHQ and LSUN experiments, we use the same evaluation setting as ADA [16].

## Appendix C    100-Shot Generation Benchmark

We collect the 100-shot datasets from the Internet. We then manually filter and crop each image as a pre-processing step. The full datasets are available here.

|  | BigGAN (baseline) | + DiffAugment (ours) |
|---|---|---|
| 100% training data | IS: 94.5   FID: 7.62 | IS: **100.8**   FID: **6.80** |
| 25% training data | IS: 46.5   FID: 25.37 | IS: **74.2**   FID: **13.28** |
|  | BigGAN (baseline) | + DiffAugment (ours) |

Figure 3: Qualitative comparison on **ImageNet** 128×128 without the truncation trick [2].

Figure 4: Qualitative comparison on **FFHQ** at 256×256 resolution with 1k, 5k, 10k, and 30k training images. Our method consistently outperforms the StyleGAN2 baselines [18] under different data percentage settings.

|  | StyleGAN2 (baseline) | + DiffAugment (ours) |
|---|---|---|
| 30k samples | FID: 10.12 | FID: **9.68** |
| 10k samples (3× fewer) | FID: 17.93 | FID: **12.07** |
| 5k samples (6× fewer) | FID: 34.69 | FID: **16.11** |
| 1k samples (30× fewer) | FID: 182.85 | FID: **42.26** |
|  | StyleGAN2 (baseline) | + DiffAugment (ours) |

Figure 5: Qualitative comparison on **LSUN-cat** at 256×256 resolution with 1k, 5k, 10k, and 30k training images. Our method consistently outperforms the StyleGAN2 baselines [18] under different data percentage settings.

Figure 6: **Style space interpolation** of our method on the 100-shot Obama, grumpy cat, panda, the Bridge of Sighs, the Medici Fountain, and the Temple of Heaven datasets without pre-training. The smooth interpolation results suggest little overfitting of our method even given small datasets.

| Query | Top-3 nearest neighbors | Query | Top-3 nearest neighbors |

Figure 7: **Nearest neighbors in pixel space** measured by the pixel-wise $L_1$ distance. Each query (on the left of the dashed lines) is a generated image of our method without pre-training (StyleGAN2 + DiffAugment) on the 100-shot or AnimalFace generation datasets. Each nearest neighbor (on the right of the dashed lines) is an original image queried from the training set with horizontal flips. The generated images are different from the training set, indicating that our model does not simply memorize the training images or overfit even given small datasets.

| Query | Top-3 nearest neighbors | Query | Top-3 nearest neighbors |

Figure 8: **Nearest neighbors in feature space** measured by the Learned Perceptual Image Patch Similarity (LPIPS) [51]. Each query (on the left of the dashed lines) is a generated image of our method without pre-training (StyleGAN2 + DiffAugment) on the 100-shot or AnimalFace generation datasets. Each nearest neighbor (on the right of the dashed lines) is an original image queried from the training set with horizontal flips. The generated images are different from the training set, indicating that our model does not simply memorize the training images or overfit even given small datasets.

| Scale/shift | MineGAN | TransferGAN | + DiffAugment | FreezeD | + DiffAugment | StyleGAN2 | + DiffAugment |
|---|---|---|---|---|---|---|---|
| FID: 54.83 | FID: 54.45 | FID: 52.61 | FID: **49.10** | FID: 47.70 | FID: **46.07** | FID: 71.71 | FID: **42.44** |

⊢——————— pre-trained with 70,000 images ———————⊣ ⊢— only **160** images —⊣

Figure 9: Qualitative comparison on the **AnimalFace-cat** [37] dataset.

| Scale/shift | MineGAN | TransferGAN | + DiffAugment | FreezeD | + DiffAugment | StyleGAN2 | + DiffAugment |
|---|---|---|---|---|---|---|---|
| FID: 83.04 | FID: 93.03 | FID: 82.38 | FID: **65.57** | FID: 70.46 | FID: **61.03** | FID: 130.19 | FID: **58.85** |

⊢——————— pre-trained with 70,000 images ———————⊣ ⊢— only **389** images —⊣

Figure 10: Qualitative comparison on the **AnimalFace-dog** [37] dataset.

| Scale/shift | MineGAN | TransferGAN | + DiffAugment | FreezeD | + DiffAugment | StyleGAN2 | + DiffAugment |
| FID: 50.72 | FID: 50.63 | FID: 48.73 | FID: **39.85** | FID: 41.87 | FID: **35.75** | FID: 80.20 | FID: **46.87** |

├────────────── pre-trained with 70,000 images ──────────────┤  ├── only **100** images ──┤

Figure 11: Qualitative comparison on the **100-shot Obama** dataset.

| Scale/shift | MineGAN | TransferGAN | + DiffAugment | FreezeD | + DiffAugment | StyleGAN2 | + DiffAugment |
| FID: 34.20 | FID: 34.54 | FID: 34.06 | FID: **29.77** | FID: 31.22 | FID: **29.34** | FID: 48.90 | FID: **27.08** |

├────────────── pre-trained with 70,000 images ──────────────┤  ├── only **100** images ──┤

Figure 12: Qualitative comparison on the **100-shot grumpy cat** dataset.

| Scale/shift | MineGAN | TransferGAN | + DiffAugment | FreezeD | + DiffAugment | StyleGAN2 | + DiffAugment |
| FID: 21.38 | FID: 14.84 | FID: 23.20 | FID: **17.12** | FID: 17.95 | FID: **14.50** | FID: 34.27 | FID: **12.06** |

├────────────── pre-trained with 70,000 images ──────────────┤  ├── only **100** images ──┤

Figure 13: Qualitative comparison on the **100-shot panda** dataset.