[Reviews · NeurIPS 2020]

Review 1

Summary and Contributions: The authors propose DiffAugment which promotes data efficiency of GANs so as to improve the effectiveness of GANs especially on limited data. The motivation of this paper is interesting and useful but the innovation might be limited.

Strengths: 1. The authors demonstrate different augmentation strategies in other tasks to improve the effectiveness of GANs. 2. The authors propose DiffAugment which transforms both the input data x and the data of generated examples G(z) to avoid discriminator overfitting and improve the effectiveness of GANs.

Weaknesses: 1. Although adopting less data to train GANs seems to be very useful, I am wondering if this can degrade the diversity of generated examples? 2. DiffAugment only adopts some current augmentation strategies, are there some augmentation strategies which are specially useful for GANs?

Correctness: The claims, method and methodology are correct.

Clarity: It is well written on the whole, but with some typos, eg. line 198: Model size natters? I think it would be better if the authors polish the paper more carefully.

Relation to Prior Work: They have summarized the prior works and discussed the difference.

Reproducibility: Yes

Additional Feedback:


Review 2

Summary and Contributions: This work proposes DiffAugment, a technique for applying standard data augmentation techniques to GANs without leaking them into the learned distribution, which is a common problem when naively applying data augmentation to the training set. The method is simple yet effective: differentiable transforms such as translation, colour jitter, or cutout are applied to both the real and fake images during calculation of the discriminator and generator loss. In this fashion the discriminator is exposed to a wider variety of input images, but the augmentations are not learned by the generator. The effectiveness of this technique is demonstrated by improving generation quality on ImageNet and CIFAR-10 datasets. Additionally, several new datasets consisting of only 100 images are introduced. It is shown that DiffAugment can be used to achieve good results in these challenging settings where standard GANs without data augmentation only produce blurry images.

Strengths: S1 - This is a simple solution to a widespread problem in GAN training. Considering the current popularity of GANs, this could easily have a very large impact. S2 - Problem and solution are well motivated. S3 - Strong experimental results, especially in the low data regime, which has been a weak spot for GANs for a while.

Weaknesses: W1 - Different datasets require different levels of data augmentation, so some grid search is required.

Correctness: Co1 - Standard GAN evaluation procedures are followed, and comparison to baselines seems fair.

Clarity: Cl1 - Paper is extremely well written and very easy to follow. I particularily like Section 3, which does a great job of describing previous attempts at data augmentation in GANs and motivating the proposed strategy.

Relation to Prior Work: RPW1 - Comparison with previous work is sufficient.

Reproducibility: Yes

Additional Feedback: AF1 - I think this is fantastic work! The paper itself was a pleasure to read, and the method is simple but quite effective. People have been trying and failing to apply data augmentation to GANs for some time, so it's great that this tool is finally available. == Post Rebuttal == After reading the other reviews and the rebuttal I will stay with the score that I originally reported. Other reviewers have highlighted some potential weaknesses of the paper that I had overlooked, but I think these small issues are outweighed by the widespread impact that this paper could have for the image generation community. I can see this work, or some future derivation of it, becoming standard for GAN training, just as data augmentation is currently standard for training any classifier for image recognition. Great work!


Review 3

Summary and Contributions: Current GAN model shows inferior performance given limited amount of data, which is due to that fact that the discriminator suffers from ovefitting, which leads to provide effective gradients to update the generator. current method to eliminate this drawback use the regularization, drop, transfer learning, data augmentation etc.. In this paper, authors consider data augmentation to alleviate the overfitting problem. The paper explores the failure when using normal data augmentation, and proposed a new one: differentiable augmentation. Specially, the data augmentation also be applied for the generated image. The gradient is allowed backpropagation via the data augmentation. The proposed method is evaluated on conditional and unconditional GAN, and show outstanding performance.

Strengths: This paper explore the overfitting problem of GAN given limited data. This paper introduces why networks ( G and D) suffers from overfitting, and proposed a new data augmentation method: differentiable augmentation. 1. Authors conduct extensive experiments to support the proposed method, including conditional, unconditional cases. 2.The paper is easy to follow. The origination is clear, and the key claim are supported by relevant experiment. 3. This paper still achieves interesting results given extremely limited images (100 human face).

Weaknesses: 1. The contribution is not enough. This paper address overfitting problem of training GAN with limited data, and proposed the differentiable augmentation. I think it is important factor, but still limited. 2. Using the accuracy (of both training and validation data) is not convincing metric, since conditional GAN suffers from limited diversity problem [1]. The generated image with limited diversity is also leveraged to train the discriminator (classifier). In this case, the learned discriminator is not suitable to evaluate the validation data. One interesting method [1] is to train one classifier using the generated images, and test it on real data. 3. I think more combination of data augmentation should be considered to support the proposed method: a new data augmentation. This paper only explore three simple terms: 'translation', 'cutout' and 'color', which fails to claim that current data augmentation fails. Some papers have investigated a series of data augmentation techniques (such as rotations, noise etc) for down stream tasks, I recommend to combine more data augmentations to support the proposed method. 4. As shown on Figure 4, are the data augmentation T (for G and D) same when updating the D? For example, at the same iteration the data augmentation T of D also is 'cutout' when G is 'cutout'. Do the data augmentation of updating D should be same or not? 5. The used the face dateset (Obama, grumpy cat and panda) have limited diversity when I check the provided data, which is easy to learn. I am doubt why the baseline result (Figure 3) is worse. 6. Table 4 reports the results of baselines. The result of MineGAN is weird. I guess author performed it by yourself since the released code of MineGAN (on StyleGAN) is later after the deadline of NeurIPS. I repeated the released code based on StyleGAN at cat and dog datasets, and got interesting results. I believe designing the miner network is little challenge, which is why the reported result is worse. 7. In figure 6, why authors consider the CIFAR-10 with 100% training data? To support the proposed method, the limited data should be consider. It would be convincing to leverage CIFAR-10 with limited data. [1] How good is my GAN? ECCV2018

Correctness: The proposed method is effective and convincing, which is evaluated by effective experiment.

Clarity: The paper is well-written, and easy to follow.

Relation to Prior Work: The contribution is clear. More papers and experiments about data augmentation should be considered.

Reproducibility: Yes

Additional Feedback: 1. Why author call 'few shot'? to my background, few-shot means that the learned model can be adapted quickly to new the class with few images. In this paper, given few images training GAN from scratch doesn't belong this definition. In fact, this paper uses 100 images, which isn't the few shot which basically has 1, 5, and 20 image for each class. --------------------- AFTER REBUTTAL --------------------- Thank authors for answering my concerns. After reading the provided explanation, I am willing to improve my score, since the proposed method addressed most of my concerns. I realize that the proposed method is a key work to use the data augmentation for GAN. This paper argues and (experimentally) shows naive data augmentation fails to lead to satisfactory result, while the proposed method (differentiable augmentation) achieve outstanding result. I guess the differentiable augmentation is important insight. Besides the proposed technique still be useful and general for the related tasks which highly rely on GAN loss, such as image-to-image translation(CycleGAN, StarGAN, DRIT, SPADE etc.) when the dataset is limited. I hope the paper will be updated based on what authors promise in rebuttal.


Review 4

Summary and Contributions: This paper proposes a method to combat overfitting of the discriminator (D) of a GAN. Previous work tackle this problem, besides regularization, by augment real and generated data and update the discriminator accordingly, however, the generator (G) is still updated with the error signals from the original generated data. The idea proposed here is to train G with the error signal backpropped through the respective differentiable augmentation function applied to the output of the G. The authors show how limited data degrades BigGAN performance measured with the FID metric and the D memorizes the training set and fails on the unseen validation data. Then the authors show how their proposed method applied to StyleGAN2 shows a much more robust behaviour against limited data compared to the unmodified GAN model. Next, different GAN augmentation strategies are compared with the proposed method where this method outperforms measured with the FID, and the D is less prone to overfitting. On the ImageNet, cifar10 and cifar100 dataset a modified BigGAN, CR-BigGAN and StyleGAN2 method outperform their original unmodified counterparts. In the Few-Shot experiments several small-sized single-category datasets were used to evaluate the few-shot performance of the method compared to several methods from literature. In both the pretrained and not pretrained setting the proposed method outperformed the competitors. Interpolation experiments in the not pretrained setting shows smooth interpolations within the target space which hints to less or no overfitting of the training data. Finally it was evaluated how model complexity based on the number of feature maps affects a modified and unmodified GAN and if a regularization hyperparameter tuning of an unmodified model could outperform the proposed augmentation variant.

Strengths: It was shown that the proposed method indeed is less prone to overfitting showing better FIDs with smaller datasizes. A comparison with different augmentation variants from literature showed superiority of the method over several augmentation functions. Experiments on three datasets, three GAN models and different datasizes show constantly better FIDs for the proposed modified versions. In the Few-Shot experiment the method shows its strength on very small single-category datasets. With 5 models from literature evaluated on 5 datasets the comparison is fair and credible. The interpolation experiments support the claim that the proposed method is a good choice for small few-shot datasets.

Weaknesses: 1) Experiments are limited to image datasets. There are GANs for sequence generation, could the proposed method help improving this models as well? In this context is there a relation with e.g. MaskGAN [1] which uses a masking as augmentation. 2) Augmentation functions have to differentiable such that the error signal can backpropagate to the generator. Which augmentation functions are differentiable? 3) The main concern about this paper is the lack of theoretical elaboration, not sure if this paper should be published at a computer vision conference. [1] MaskGAN: Better Text Generation via Filling in the ____, William Fedus and Ian Goodfellow and Andrew Dai, 2018, URL= https://openreview.net/pdf?id=ByOExmWAb, ICLR 2018

Correctness: Claims and method look correct as well as the empirical methodology.

Clarity: The paper is well written and easy to understand.

Relation to Prior Work: Short answer yes.

Reproducibility: Yes

Additional Feedback:

[Author Response · NeurIPS 2020]

**Technical Contributions (R3, R4).** Augmentation plays a key role in many machine learning systems. This paper has addressed discriminator overfitting — a fundamental problem in the GAN literature — via Differentiable Augmentation, which boosts GANs not only with **limited data** but also on the **large-scale** datasets. We believe that this is an important algorithmic contribution to the ML community. Though not theoretical contribution, many well-known empirical papers were also published at ML conferences (*e.g.*, DCGAN in ICLR, and LaplacianGAN in NeurIPS).

**Image Diversity (R1).** Our method can improve diversity. As suggested by R1, we use the recall metric [1] that estimates the coverage of the generated distribution and hence reflects the diversity. Ours with **20%** CIFAR-10 data (recall: **0.39**) is higher than the StyleGAN2 baseline with **20%** data (recall: **0.24**), even higher than its with **100%** data (recall: **0.33**). We will include it in our revision.

**Clarification of $T$ across $G$ and $D$ (R3).** $T$ is required to be the same random function but not necessarily the same random seed across $G$ and $D$, since $G$ and $D$ are updated in different forward-backward iterations. We will clarify this in the revision.

**Results of MineGAN (R3).** We rerun MineGAN using their newly released code and will update their FID (obama: 50.63; grumpy cat: 34.54; panda: 14.84; cat: 54.45; dog: 93.03) and figures in the revision. Ours (StyleGAN2 + DiffAugment) outperforms MineGAN on 4 out of 5 datasets.

**Why 100% Data in Fig. 6? (R3)** We show that DiffAugment **even works** with 100% of CIFAR-10 data, where the discriminator still severely overfits the training set. This phenomenon is more severe with limited data. As the reviewer requests, with 10% CIFAR-10 data, at 10k iterations when the BigGAN baseline collapses, its $D$'s training/validation accuracy is 99%/18% (**81%** difference, **severe** overfitting), while ours is 90%/41% (**49%** difference, **less** overfitting). Ours continues stable training for over 60k iterations, considerably alleviating the overfitting problem.

**Naming of "Few-Shot" (R3).** Thanks and we are happy to change it to "100-shot" in the revision.

**Application to NLP (R4).** Like image inpainting, the masking process in MaskGAN as R4 mentioned is used to construct the conditional input. This is not a form of discriminator augmentation, as $D$ is still seeing the unmasked training set. We will cite MaskGAN and discuss its connection to our method. DiffAugment for NLP tasks is an interesting direction. We leave this for future work.

**Choices of Augmentations (R1, R3).** We mainly investigate the algorithmic perspective — where and how to apply augmentations to GANs; exhausting the set of augmentations is beyond the scope of this paper. In fact, we have tried many other augmentations like random scaling, rotations, shearing, smoothing, sharpening, and Gaussian noise but did not find them helpful. Moreover, when they are applied as "Augment reals only" or "Augment $D$ only", all the results are consistently worse than the baseline. Thus we find that *Color*, *Translation*, and *Cutout* are especially effective for GANs. The simplicity also makes it easier to be deployed. We will discuss other augmentations in the revision.

**Which Augmentations are Differentiable? (R4)** Most existing augmentations could have a differentiable implementation, but they are currently absent in the widely used TensorFlow or PyTorch. Our code release provides differentiable implementations, which would benefit the community.

**Grid Search (R2).** Grid search can further optimize the performance, but our fixed *Color + Translation + Cutout* DiffAugment already works fairly well in most limited data settings, including CIFAR, few-shot, and our new results in Table 1. Although we used *Translation + Cutout* for the BigGAN models in the CIFAR tables, we later find that they can be further improved if *Color* is used as well (*e.g.*, from FID: 22 to 20 with 10% CIFAR-10 data). This combination is especially effective for GANs. Besides, we did not tune the level of each individual augmentation, which we found little beneficial, so the search space is significantly reduced.

| Dataset: | FFHQ 256×256 | | | | LSUN-Cat 256×256 | | | |
|---|---|---|---|---|---|---|---|---|
| # Training samples: | 30k | 10k | 5k | 1k | 30k | 10k | 5k | 1k |
| StyleGAN2 | 6.16 | 14.75 | 26.60 | 62.16 | 10.12 | 17.93 | 34.69 | 182.85 |
| + DiffAugment | **5.05** | **7.86** | **10.45** | **25.66** | **9.68** | **12.07** | **16.11** | **42.26** |

Figure 1: FFHQ and LSUN-Cat results with the fixed *Color + Translation + Cutout* DiffAugment.

**Results of Baseline for 100-Shot Generation (R3).** Although each of our datasets contains only 1 object, their facial expressions, backgrounds, and poses are fairly diverse. $D$ can easily memorize all those 100 training images and that's why the baseline StyleGAN2 is poor. What Fig. 3 presents is already the **best** training snapshot of the baseline model. It can be even worse if the training is longer.

**Metrics for Overfitting (R3).** As suggested by R3, the GAN-train/GAN-test metric is a good metric for assessing the generated images of the **generator**. E.g., the GAN-train/GAN-test of the Big-GAN baseline with 10% data is 53.1%/72.4%, while ours achieves significantly better 62.7%/80.9%. However, in this paper, we only use the discriminator's accuracy on the real training/validation set to see if the **discriminator** overfits the real images. We will clarify this in the revision.

**Typos (R1).** Thanks for the suggestion. We will revise the paper thoroughly.

[1] Kynkäänniemi et al. Improved precision and recall metric for assessing generative models. In *Advances in Neural Information Processing Systems*, 2019. 1



[Meta-Review · NeurIPS 2020]

All four reviewers support acceptance. AC agrees and thinks that this is a timely work and clear contribution (with solid experiments) in the GAN literature. Hence, AC recommends acceptance with no doubt.